# ParK: Sound and Efficient Kernel Ridge Regression by Feature Space Partitions

**Luigi Carratino**\*
MaLGa - DIBRIS, University of Genova
`luigi.carratino@dibris.unige.it`

**Stefano Vigogna**\*
MaLGa - DIBRIS, University of Genova
`vigogna@dibris.unige.it`

**Daniele Calandriello**
DeepMind Paris
`dcalandriello@google.com`

**Lorenzo Rosasco**
MaLGa - DIBRIS, University of Genova
IIT, CBMM - MIT
`lrosasco@mit.edu`

## Abstract

We introduce ParK, a new large-scale solver for kernel ridge regression. Our approach combines partitioning with random projections and iterative optimization to reduce space and time complexity while provably maintaining the same statistical accuracy. In particular, constructing suitable partitions directly in the feature space rather than in the input space, we promote orthogonality between the local estimators, thus ensuring that key quantities such as local effective dimension and bias remain under control. We characterize the statistical-computational tradeoff of our model, and demonstrate the effectiveness of our method by numerical experiments on large-scale datasets.

## 1 Introduction

The development of provably accurate and efficient algorithms for learning is key to tackle modern large-scale applications. Kernel methods [31, 32] provide a natural ground to develop this research direction. On the one hand they have sound statistical guarantees [7, 32, 33], but on the other hand their basic implementations are limited to sample size of only a few tens of thousands of points [32, Chapter 11]. Recent years have witnessed a growing literature introducing algorithmic solutions to improve efficiency, but also theoretical guarantees that quantify how accuracy is affected.

We next recall a few lines of work relevant to our study. A first line of work is based on exploiting ideas from optimization and numerical analysis. This includes for example gradient methods [37], as well as their accelerated [3], stochastic [10], preconditioned [13] and distributed [27] variants. A second line of work is based on using randomized approaches to reduce the size of the problem to be solved. This includes Nyström approximations [36], random features [25] and more generally sketching methods [1]. The theoretical properties of these methods have been recently characterized in terms of sharp statistical bounds [28, 30]. Finally, a third line of work considers different partitioning strategies to divide the estimation step in smaller subproblems. This approach is based on splitting the input space in regions where local estimators are defined [23, 35, 34, 11, 24, 5]. In this context, the emphasis is typically on allowing the estimation of larger classes of functions. Another form of partitioning, called divide-and-conquer, is instead based on randomly splitting the training data to then obtain a global estimator by averaging [38, 20, 14]. In this approach, the focus is primarily on computational saving. Notably, a number of works have considered combinations of these ideas, see for example [6, 29, 8, 24, 19].

---

\*equal contribution

35th Conference on Neural Information Processing Systems (NeurIPS 2021).

In this paper we propose and study a local kernel algorithm, called ParK, combining partitioning with iterative optimization and sketching. Our goal is to provide an efficient and accurate approximation to a global kernel ridge regression estimator. The main novelty in ParK is in the form of the considered partition, that we define in the feature space, rather than in the input space as in traditional partitioning methods. This allows to promote orthogonality between the local estimators, and thus to control the local effective dimension and the local bias. Given a partition, local kernel ridge estimators are computed using sketching and preconditioned conjugate gradient iterations [29]. From a theoretical point of view, our main contribution is characterizing the statistical properties of ParK, in terms of conditions on the partition and the choice of the hyper-parameters. Borrowing ideas from subspace clustering [12], we show that the minimal angle between suitable subspaces induced by the partition plays a crucial role. Indeed, our analysis shows that, if such an angle is sufficiently large, ParK can achieve the same accuracy as global kernel ridge regression estimators, with only a fraction of computations. Our theoretical results are complemented with numerical experiments on very large datasets, which show that ParK can indeed provide excellent performances, on par and often better than the best available large-scale kernel methods.

The rest of the paper is organized as follows. In Section 2 we state the problem and recall the basics of kernel ridge regression. In Section 3 we illustrate our algorithm. In Section 4 we analyze the prediction error of our method. In Section 5 we present the results of our numerical experiments. In Section 6 we draw some conclusions and report the main limitations of our work. Additional proofs and details are collected in Appendix A.

## 2   Background

Let $(x_i, y_i)$ with $i \in [n] = \{1, \ldots, n\}$ be $n$ pairs of points in $\mathcal{X} \times \mathcal{Y}$, where $\mathcal{X} \subseteq \mathbb{R}^d$ with $d \in \mathbb{N}$ and $\mathcal{Y} \subseteq \mathbb{R}$. We assume the relation between input points $x_i$ and output points $y_i$ to be determined by the noisy evaluations of an unknown function $f_* : \mathcal{X} \to \mathcal{Y}$ as

$$y_i = f_*(x_i) + \varepsilon_i \qquad i \in [n]. \tag{1}$$

Based on the samples $(x_i, y_i)$, we want to estimate the function $f_*$, searching for solutions in a suitable hypothesis space $\mathcal{H}$ as detailed below.

Let $\mathcal{H}$ be a reproducing kernel Hilbert space (RKHS), that is, a Hilbert space of functions with inner product $\langle \cdot, \cdot \rangle_{\mathcal{H}}$ and symmetric positive definite kernel $K : \mathcal{X} \times \mathcal{X} \to \mathbb{R}$ such that $K_x = K(x, \cdot) \in \mathcal{H}$ and $f(x) = \langle f, K_x \rangle_{\mathcal{H}}$ for all $f \in \mathcal{H}, x \in \mathcal{X}$. We recall that, for every RKHS $\mathcal{H}$, there exist a Hilbert feature space $\mathcal{F}$ and a feature map $\phi : \mathcal{X} \to \mathcal{F}$ such that $K(x, x') = \langle \phi(x), \phi(x') \rangle_{\mathcal{F}}$ for all $x, x' \in \mathcal{X}$. The feature map is not unique; in particular, one may take, as we do in all that follows, $\mathcal{F} = \mathcal{H}$ and $\phi(x) = K_x$, in which case $\mathcal{H} = \overline{\mathrm{span}}\, \phi(\mathcal{X})$, where $\phi(\mathcal{X}) = \{\phi(x) : x \in \mathcal{X}\}$.

Kernel ridge regression (KRR) corresponds to minimizing

$$\min_{f \in \mathcal{H}} \frac{1}{n} \sum_{i=1}^{n} |f(x_i) - y_i|^2 + \lambda \|f\|_{\mathcal{H}}^2, \tag{2}$$

where $\lambda > 0$ and $\|f\|_{\mathcal{H}}^2 = \langle f, f \rangle_{\mathcal{H}}$. By the representer theorem [31], the (unique) solution to problem (2) can be written as

$$\widehat{f}_\lambda(x) = \sum_{i=1}^{n} \alpha_i K(x_i, x), \qquad \alpha = (K_n + \lambda n I)^{-1} Y, \tag{3}$$

where $\alpha = [\alpha_1, \ldots, \alpha_n]^\top, Y = [y_1, \ldots, y_n]^\top \in \mathbb{R}^n$, and $K_n \in \mathbb{R}^{n \times n}$ is the kernel matrix defined by $(K_n)_{i,j} = K(x_i, x_j)$ for $i, j \in [n]$. As a consequence, the estimator (3) can be derived restricting the minimization problem (2) to the finite-dimensional subspace $\mathcal{H}_n = \mathrm{span}\{\phi(x_i) \mid i \in [n]\}$. Computing (3) for large $n$ is prohibitively expensive, as space and time complexities are, respectively, $O(n^2)$ and $O(n^3)$. The goal of this paper is to provide an algorithm to compute an efficient approximation to (3).

## 3   Algorithm

Our method combines diverse techniques, including partitioning, sketching and preconditioned iterative optimization. We begin focusing on partitioning. While classical partitioning methods

construct partitions in the input space, the main novelty of our approach is that we construct partitions in the feature space. Note that, in the case of a universal kernel on a compact input space, every feature map is injective [32, Lemma 4.55], hence every partition of the input space defines a corresponding partition of the feature space. Thus, we may see our approach as a generalization of classical input space partitioning approaches. As will become apparent from our analysis, the performance of a partitioned kernel estimator depends crucially on two main quantities: the local biases and the local effective dimensions. Since both quantities are strictly related to the RKHS of choice, constructing partitions in feature space allows for a more direct control. In particular, promoting orthogonality in the RKHS metric will generate feature space partitions which tend to minimize both the local biases and the local effective dimensions. In the next section we start discussing how such partitions can be defined.

## 3.1 Learning on feature space partitions

For $Q \in \mathbb{N}$, we define a partition of $\phi(\mathcal{X})$ as a family $\{V_q\}_{q \in [Q]}$ of subsets $V_q \subseteq \phi(\mathcal{X})$ such that

$$\phi(\mathcal{X}) = \bigcup_{q=1}^{Q} V_q \qquad V_q \cap V_k = \varnothing \quad q \neq k. \tag{4}$$

The partition (4) induces a local subsampling of the training set and a local hypothesis space. Namely, we define

$$[n]_q = \{i \in [n] : \phi(x_i) \in V_q\}, \qquad \mathcal{H}_q = \overline{\mathrm{span}}\{V_q\}.$$

Also, we denote by $n_q = \#[n]_q$ the local subsampling rate.

**Voronoi partitions.** Notice that so far $V_q$ is an arbitrary subset of $\mathcal{H}$, and therefore computing the set $[n]_q$ can be arbitrarily difficult (e.g., $V_q$ could be defined using an infinite number of constraints and be non-computable). For this reason, although our theoretical analysis holds for any partition defined as in (4), we focus on the special case of Voronoi partitions, where the subsets (also called cells) are induced by a set of $Q$ centroids $\{\phi(c_q)\}_{q=1}^{Q}$ with $c_q \in \mathcal{X}$ points in the input space. Then, each cell $V_q$ is uniquely defined as

$$V_q = \{\phi(x) : q = \arg\min_{k \in [Q]} \|\phi(x) - \phi(c_k)\|_{\mathcal{H}}^2\},$$

with ties broken arbitrarily (e.g., by assigning the point to the cell with the smaller $q$). It is now possible to identify the set of indices $[n]_q$ using the RKHS distance

$$\|\phi(x) - \phi(x')\|_{\mathcal{H}}^2 = K(x, x) + K(x', x') - 2K(x, x'), \tag{5}$$

computing the distance to each centroid and taking the minimum.

We remark that our approach based on directly partitioning the feature space has quite different implications compared to previous approaches that partition the input space. For example, a Voronoi partition of the feature space is very different from a Voronoi partition of the input space, since the pre-image $\{x \in \mathcal{X} : \phi(x) \in V_q\}$ does not need to follow any Voronoi shape. Moving from input to feature space partitions also opens new computational challenges. For example, we choose to explicitly represent the cell centroid as $\phi(c_q)$ so that computing the distance and the assignment of each point to a centroid is a $O(1)$ operation. If instead we chose a more complex centroid, such as a cluster barycenter generated by kernel $k$-means, or an eigenvector computed by kernel PCA, this complexity might be much larger. As an example, the barycenter of a cluster of $m$ points in $\mathcal{H}$ might not correspond to any single point in $\mathcal{X}$, and therefore cannot be explicitly represented, but only implicitly as an average of $m$ points in $\mathcal{H}$. Therefore, computing a distance to such a centroid would be an $O(m)$ operation rather than a $O(1)$. These and more subtle pitfalls appear only when we consider the more flexible framework of feature space partitions.

**Minimal principal angle.** Focusing on partitions of Voronoi type, constructing a good partition is equivalent to choosing a set of centroids that preserve the learning accuracy as much as possible. As we rigorously show in Section 4, this can be guaranteed by choosing centroids that maximize the minimal principal angle between subspaces. This quantity frequently appears in the analysis of

subspace clustering [12], and will be important for us to control both the bias and the variance of our estimator. The first principal angle between two linear subspaces $U$ and $W$ of a Hilbert space of inner product $\langle \cdot, \cdot \rangle$ and norm $\| \cdot \|$ is defined as

$$\angle(U, W) = \min\{\arccos(\langle u, w \rangle) : u \in U, w \in W, \|u\| = \|v\| = 1\}.$$

We call $\theta$ the minimal first principal angle between the subspaces $\mathcal{H}_q$, that is,

$$\theta = \min_{q \neq k} \angle(\mathcal{H}_q, \mathcal{H}_k). \tag{6}$$

Once again, for computational reasons we cannot use direct optimization of this quantity in $\mathcal{H}$ as our objective, since the optimal centroid placement might be impossible to express using points from $\mathcal{X}$. Instead, to promote large principal angles and obtain centroids that are computationally easy to handle, we consider the following greedy iterative procedure. Let $X = \{x_i : i \in [n]\}$. Then

$$c_1 = \arg\max_{c \in X} K(c, c), \qquad c_{q+1} = \arg\max_{c \in X \setminus \{c_1, \dots c_q\}} \mathrm{SC}_q(c), \tag{7}$$

where

$$\mathrm{SC}_q(c) = K(c, c) - [K(c, c_1), \dots, K(c, c_q)]^\top K_q^{-1} [K(c, c_1), \dots, K(c, c_q)]$$

is the Schur complement of a new candidate centroid $c$ with respect to the $q$ already selected centroids $\{c_1, \dots, c_q\}$, and $K_q \in \mathbb{R}^{q \times q}$ is defined by $(K_q)_{i,j} = K(c_i, c_j)$. Note that the inversion of $K_q$ can be efficiently computed using rank-1 updates. This strategy has been originally proposed by [9] with the goal of maximizing the volume spanned by the points in the feature space, which is achieved when the angle between all points selected is large as required by our condition. Crucially, it is also easy to apply to RKHS's, since computing Schur complements involves only inner products. Beyond promoting large volume and orthogonality, the Schur complement also has important links with uncertainty estimation and spectral approximation. In particular, $\mathrm{SC}_q(c)$ is also equivalent to the posterior variance of $c$ in a Gaussian process [26], and to the leverage score of $c$ w.r.t. the already selected point in the context of randomized linear algebra [21].

## 3.2 Learning local KRR estimators by sketched preconditioned conjugate gradient

For each cell of a partition, a local estimator $\widehat{f}_q$ can be defined as the solution to the local KRR problem

$$\min_{f \in \mathcal{H}_q} \frac{1}{n_q} \sum_{i \in [n]_q} |f(x_i) - y_i|^2 + \lambda_q \|f\|_{\mathcal{H}}^2 \tag{8}$$

with $\lambda_q > 0$. Given the local estimators $\widehat{f}_q$, we then define a global estimator $\overline{f}$ by

$$\overline{f}(x) = \widehat{f}_q(x) \qquad \text{if } \phi(x) \in V_q. \tag{9}$$

Note that the evaluation of the global estimator at a point needs only one local estimator.

Guidance on how to pick the values $\lambda_q$ in (8) will follow from our theoretical analysis. Meanwhile, we focus on how to efficiently solve the minimization problems (8). Let $X_q = \{x_i \in X : i \in [n]_q\} \in \mathbb{R}^{n_q \times d}$ and $Y_q = \{y_i \in Y : i \in [n]_q\} \in \mathbb{R}^{n_q}$ be the local subsets of input/output points, and let $K_{n_q} \in \mathbb{R}^{n_q \times n_q}$ be the local kernel matrix with entries $(K_{n_q})_{i,j} = K(x_i, x_j)$ for $i, j \in [n]_q$. Following the same ideas to derive (3), one could compute $\widehat{f}_q$ by

$$\widehat{f}_q(x) = \sum_{i \in [n]_q} (\alpha_q)_i K(x_i, x), \qquad \alpha_q = \left(K_{n_q} + \lambda_q n_q I\right)^{-1} Y_q. \tag{10}$$

This would already result in a smaller computational burden compared to the vanilla KRR estimator (3): the space and time complexities are now $O(\max_{q \in [Q]} n_q^2)$ and $O(\sum_{q \in [Q]} n_q^3)$, potentially with $n_q \ll n$. Moreover, an additional saving in time can be obtained by distributing each task (10) over $Q$ different machines, leading to $O(\max_{q \in [Q]} n_q^3)$ time complexity. However, the scaling in $n_q$ is still quadratic and cubical. To improve these dependencies, we solve (10) only approximately, using the FALKON algorithm proposed in [29]. To this end, we first need to introduce several key ingredients. While the following constructions hold in general for any set of points, here we adapt them to the partition setting outlined in the previous section.

**Local Nyström subsampling.** For each $q \in [Q]$, we consider a subset of $m_q \leq n_q$ points

$$\widetilde{X}_q = \{\widetilde{x}_{q,1}, \ldots, \widetilde{x}_{q,m_q}\} \subseteq X_q \tag{11}$$

sampled uniformly at random from $X_q$. We then define $K_{m_q} \in \mathbb{R}^{m_q \times m_q}$ by $(K_{m_q})_{i,j} = K(\widetilde{x}_{q,i}, \widetilde{x}_{q,j})$ for $i, j \in [m_q]$, and $K_{n_q m_q} \in \mathbb{R}^{n_q \times m_q}$ by $(K_{n_q m_q})_{i,j} = K(x_i, \widetilde{x}_{q,j})$ for $i \in [n]_q, j \in [m_q]$.

**Local Preconditioner.** For each $q \in [Q]$, we define the local (sketched) preconditioner $B_q \in \mathbb{R}^{m_q \times m_q}$ as

$$B_q B_q^\top = (\frac{n_q}{m_q} K_{m_q}^2 + \lambda_q n_q K_{m_q})^{-1}.$$

**Conjugate gradient descent.** We let $\widetilde{\beta}_{q,t} \in \mathbb{R}^{m_q}$ be the $t$-th iteration of conjugate gradient minimizing

$$\mathcal{L}_q(\beta) = \frac{1}{n_q} \|K_{n_q m_q} B_q \beta - Y_q\|^2 + \lambda_q \beta^\top (B_q^\top K_{m_q} B_q) \beta. \tag{12}$$

Finally, we define the local FALKON estimator

$$\widetilde{f}_{q,t}(x) = \sum_{i=1}^{m_q} (B_q \widetilde{\beta}_{q,t})_i K(\widetilde{x}_i, x) \qquad q \in [Q]. \tag{13}$$

### 3.3 ParK

We are now ready to present ParK. Let $(\phi(c_q))_{q=1}^Q$ with $c_q \in X$ be the centroids of the cells $(V_q)_{q=1}^Q$ selected greedily according to (7). We define the ParK estimator as

$$\overline{f}_t(x) = \widetilde{f}_{q,t}(x) \qquad \text{if } \phi(x) \in V_q. \tag{14}$$

The algorithm to train the above estimator (see Algorithm 1) consists of three main parts. The first one greedily identifies the representative points $(c_q)_{q=1}^Q$ such that $(\phi(c_q))_{q=1}^Q$ are the centroids of the cells; the second one identifies the subsets of points $X_q, Y_q$ associated to each cell; the third one uses the FALKON algorithm to solve the local minimization problem (12) for each $X_q, Y_q$ with $q \in [Q]$, thus deriving the $Q$ local estimators (13). At prediction time, the algorithm first identifies to which cell the test point belongs, and then proceeds using the local estimator of the selected cell to predict the output (see Algorithm 2). Note that the RKHS distances in lIne 4 of Algorithm 1 and line 1 of Algorithm 2) are computed using the polarization identity (5).

---

**Algorithm 1** ParK: Train

---

**Require:** Training set $X = (x_i)_{i=1}^n \in \mathbb{R}^{n \times d}, Y = (y_i)_{i=1}^n \in \mathbb{R}^n$, numbers of local Nyström centers $\{m_q\}_{q=1}^Q \in \mathbb{N}^Q$, local regularization parameters $\{\lambda_q\}_{q=1}^Q \in \mathbb{R}_+^Q$, number of local iterations $\{t_q\}_{q=1}^Q \in \mathbb{N}^Q$.
1: Initialize $[n]_q = \{\}$ for all $q \in [Q]$
2: Greedily select $(\phi(c_q))_{q=1}^Q$ according to (7)
3: **for** $i = 1, \ldots, n$ **do**
4:     Compute $\overline{q} = \text{argmin}_{q \in [Q]} \|\phi(x_i) - \phi(c_q)\|_{\mathcal{H}}^2$
5:     Update $[n]_{\overline{q}} = [n]_{\overline{q}} \cup \{i\}$
6: **end for**
7: **for** $q = 1, \ldots, Q$ **do**
8:     Select $X_q = \{x_i \in X : i \in [n]_q\}$ and $Y_q = \{y_i \in Y : i \in [n]_q\}$
9:     Compute $\widetilde{f}_{q,t_q}$ as in eq. (13) using $X_q, Y_q$
10: **end for**
11: Collect the local estimators $\widetilde{f}_{q,t_q}$ and return the ParK estimator $\overline{f}_t$ as in eq. (14)

---

---

**Algorithm 2** ParK: Predict

---

**Require:** Test point $x \in \mathbb{R}^d$, local estimators $\{\widetilde{f}_{q,t}\}_{q=1}^Q$, representatives of partition $C = (c_q)_{q=1}^Q$
  1: Select $\overline{q} = \operatorname{argmin}_{q \in [Q]} \|\phi(x) - \phi(c_q)\|_{\mathcal{H}}^2$
  2: Evaluate $\widetilde{f}_{\overline{q},t}(x)$

---

The time complexity of training ParK is $O(Q^2 n \log(n))$ to compute the centroids, $O(Qn)$ to compute the indices $[n]_q$, and $O(t_q m_q n_q)$ to compute each local estimator. Putting these quantities together we get $O(Q^2 n \log(n) + \sum_{q \in [Q]} t_q m_q n_q)$ in time, and $O(\max_{q \in [Q]} m_q{}^2)$ in space. If we parallelize the training of the local estimators over $Q$ machines, the time complexity further reduces to $O(Q^2 n \log(n) + \max_{q \in [Q]} t_q m_q n_q)$. In many practical scenarios, we can think $Q$ as $O(1)$. For example, in all our experiments we take $Q = 32$ (see Section 5). We compare the complexity of several KRR solver in Table 1.

Table 1: Computational complexity of some KRR solvers (up to constants). For D&C and ParK, we report the time complexity on $Q$ parallel machines and the space requirement for each machine.

| | naive | iterative [37] | Nyström/RF [36, 25] | FALKON [29] | D&C [38] | ParK |
|---|---|---|---|---|---|---|
| space | $n^2$ | $n^2$ | $m^2$ | $m^2$ | $(n/Q)^2$ | $\max_q m_q{}^2$ |
| time | $n^3$ | $tn^2$ | $m^2n$ | $tmn$ | $(n/Q)^3$ | $Q^2 n \log(n) + \max_q t_q m_q n_q$ |
| test | $n$ | $n$ | $m$ | $m$ | $n$ | $Q + \max_q m_q$ |

### Space partitioning vs data splitting

We conclude this section commenting on a different yet related distributed approach. As briefly recalled in the introduction, a straightforward way to decompose the KRR problem is by a simple split of the training data. For example, one can divide the samples uniformly at random into $Q$ disjoint subsets of cardinality $n_q = n/Q$. Methods performing such a step are known as divide-and-conquer [38, 20, 14]. Consisting essentially in a block diagonal approximation of the kernel matrix, the resulting final estimator is an average of globally subsampled models. Divide-and-conquer methods are appealing due to the extreme simplicity of the splitting procedure and the direct control of the subsampling rates $n_q$. However, they can suffer from worse approximation error (see discussion in [34]), and be expensive at test and evaluation time. On the other hand, partitions present several potential benefits. First, data splitting is a byproduct of a geometric partition. This opens to the opportunity of exploiting the structure of the space, for instance enforcing notions of locality or orthogonality. Consistently, the final estimator is a union of local estimators, as opposed to an average of global ones. Hence, partitioning may enhance the approximation power of the model, capturing relevant local correlations [23]. As another consequence, at evaluation time only one local estimator, instead of the average of all estimators, needs to be called, yielding further computational saving. These nice properties have motivated a fruitful line of research, notably [23, 34, 24], where the advantage in the partitioning approach has been studied both in statistical and in computational terms. In this paper we concentrate on the computational aspects, expanding on theoretical tradeoffs outlined in [34, 24] and developing [24] with new algorithmic ideas.

## 4 Theory

To simplify the analysis and better highlight the new ideas in play, we consider the problem (1) in a fixed design setting [2, 15], where the $x_i$ are deterministic and the $\varepsilon_i$ are independent and identically distributed random variables.

Let $L^2 = L^2(\rho)$ with $\rho = \frac{1}{n} \sum_{i=1}^n \delta_{x_i}$. We may identify $L^2$ with $\mathbb{R}^n$ endowed with the inner product $\langle u, w \rangle_{L^2} = \frac{1}{n} u^\top w$. We define the excess risk of an estimate $\widehat{f}$ of $f_*$ in problem (1) as

$$\mathcal{R}(\widehat{f}) = \|\widehat{f} - f_*\|_{L^2}^2 = \frac{1}{n} \sum_{i=1}^n |\widehat{f}(x_i) - f_*(x_i)|^2. \tag{15}$$

We are interested in studying the performance of the estimator $\overline{f}_t$ defined in (14) given a partition (4), as measured by the excess risk (15). Our theory will suggest how to construct the partition and tune the regularization in order to get the best learning rate.

## 4.1 Definitions and assumptions

We start by defining some relevant operators in global and local variants. In view of (3) (and the fixed design setting), we assume without loss of generality that $\mathcal{H} = \mathcal{H}_n$. We define the covariance operator $T : \mathcal{H} \to \mathcal{H}$ as $T = \frac{1}{n} \sum_{i \in [n]} \phi(x_i) \otimes \phi(x_i)$, where, for $v, w \in \mathcal{H}$, $v \otimes w$ denotes the operator $u \in \mathcal{H} \mapsto \langle u, v \rangle_{\mathcal{H}} w \in \mathcal{H}$. The operator $T$ is standard in the analysis of kernel methods [7]. We now define the local version of the covariance operator conditioned on the partitioning (4). Thanks to (10) (and the fixed design setting), we can assume without loss of generality that $\mathcal{H}_q = \mathrm{span}\{\phi(x_i) : i \in [n]_q\}$. The local covariance operator $T_q : \mathcal{H} \to \mathcal{H}$ is defined as $T_q = \frac{1}{n_q} \sum_{i \in [n]_q} \phi(x_i) \otimes \phi(x_i)$. We denote with $P_q : \mathcal{H} \to \mathcal{H}$ the orthogonal projection onto the subspace $\mathcal{H}_q$. For all $q \in [Q]$, we let $\rho_q = n_q/n$. Recall that we denote by $\theta$ the minimal principal angle between the subspaces $\mathcal{H}_q$, as defined in (6).

To measure the capacity of the hypothesis spaces, we will use the standard notion of effective dimension [7].

**Effective dimension.** The (global) effective dimension of the space $\mathcal{H}$ is given by

$$\mathcal{N}(\lambda) = \mathrm{Tr}((T + \lambda)^{-1} T) \qquad \lambda > 0.$$

Consistently, we define the local effective dimension of each space $\mathcal{H}_q$ as

$$\mathcal{N}_q(\lambda_q) = \mathrm{Tr}((T_q + \lambda_q)^{-1} T_q) \qquad \lambda_q > 0.$$

We also define the local maximal degrees of freedom [2] as

$$\mathcal{N}_{\infty,q}(\lambda_q) = \sup_{x \in X_q} \langle \phi(x), (T_q + \lambda_q)^{-1} \phi(x) \rangle_{\mathcal{H}} \qquad \lambda_q > 0,$$

which gives the bound $\mathcal{N}_q(\lambda_q) \leq \mathcal{N}_{\infty,q}(\lambda_q) \leq \lambda_q^{-1} \sup_x K(x, x)$. The effective dimension is related to the spectrum decay of the covariance operator, and thus it provides a way to quantify how many important eigenfunctions the RKHS contains. In this sense, it serves as an implicit number of parameters for the nonparametric model represented by the RKHS. The interplay between global and local effective dimensions, hence between global and local model complexity, will play a major role in our analysis. Similarly, there exist an interplay between a local and a global version of the maximal degrees of freedom $\mathcal{N}_{\infty}(\lambda)$, which is also connected to the coherence of the $T$ operator, and to the concept of maximal leverage score [2].

We will need a few basic assumptions.

**Assumption 1.** $f_* \in \mathcal{H}$.

**Assumption 2.** $\kappa^2 = \sup_{x \in \mathcal{X}} K(x, x) < \infty$.

**Assumption 3.** *The noise variables $\varepsilon_i$ are i.i.d. sub-Gaussian of variance proxy $\sigma^2 < \infty$, $i \in [n]$.*

Assumption 3 is standard in the analysis of any regression model. In particular, sub-Gaussianity allows to control the tails of the noise, and therefore to establish bounds in high probability. Bounded and Gaussian noise are examples, but any variable with sub-Gaussian tail is covered. Assumptions 1 and 2 are instead typical of kernel methods. With Assumption 1, we suppose that the RKHS is a well specified model. We stick to Assumption 1 for simplicity, but we could easily relax it assuming the existence of a function in the RKHS with same excess risk as $f_*$, or considering the excess risk with respect to the best in class. Assumption 2 allows to provide explicit bound for kernel related quantities, and ensures in particular that functions in the RKHS are bounded.

## 4.2 Main results

Our first proposition generalizes the classical bias-variance tradeoff of KRR estimators incorporating iterative optimization, random projections and feature partitioning. The result is a high probability bound for the excess risk of our ParK estimator.

**Proposition 1.** *Let $\delta \in (0,1)$. Under the regression model* (1) *and the assumptions of Section 4.1, let $\overline{f}_{t_q}$ be the ParK estimator as defined in* (14). *If for each $q \in [Q]$, $0 < \lambda_q \leq \kappa^2$,*

$$m_q \geq 5[1 + 14\mathcal{N}_{\infty,q}(\lambda_q)]\log(\frac{8\kappa^2}{\lambda_q\delta}), \qquad t_q \geq 2\log\left(4\sigma^2\left(\|P_q f_*\|_{\mathcal{H}}^2\lambda_q\right)^{-1/2}\right),$$

*then, with probability at least $1 - 4\delta$,*

$$\mathcal{R}(\overline{f}_{t_q}) \leq 16\sum_{q=1}^{Q}\|P_q f_*\|_{\mathcal{H}}^2\lambda_q\rho_q + \sigma^2\sum_{q=1}^{Q}\frac{\mathcal{N}_q(\lambda_q) + \sqrt{\mathcal{N}_q(\lambda_q)\log(1/\delta)} + 2\log(1/\delta)}{n}.$$

The proof of Proposition 1 is given in Appendix A.2. The bound consists of a bias and a variance term. The bias term is an average of local biases, measured by the projection of the target function onto the local hypothesis spaces, regularized by a local penalization. The variance term is essentially the ratio between the sum of local effective dimensions and the global sample size. We are going to control bias and variance in the next two propositions, whose proof is postponed to Appendix A.3. For the bias, we prove the following generalized Bessel inequality.

**Proposition 2.** *With the definitions of Section 4.1, we have*

$$\sum_{q=1}^{Q}\|P_q f_*\|_{\mathcal{H}}^2 \leq (1 + Q^2\cos(\theta))\|f_*\|_{\mathcal{H}}^2.$$

Proposition 2 bounds the possible redundancy of the local projections by the minimal principal angle between the local subspaces. In particular, if the local subspaces are an orthogonal decomposition of the global space, the partitioned estimator has no additional local bias. On the other hand, lack of orthogonality results in a larger bias. Turning to the variance, we obtain the following bound on the local effective dimensions.

**Proposition 3.** *With the definitions of Section 4.1, for $\lambda_q = \lambda\rho_q^{-1}$ we have*

$$\sum_{q=1}^{Q}\mathcal{N}_q(\lambda_q) \leq \left(1 + \kappa^2\frac{\cos^2(\theta)}{\lambda}\right)\mathcal{N}(\lambda).$$

Once again, the minimal principal angle controls the ratio between local and global quantities. Where there is perfect orthogonality, splitting the hypothesis space does not increase the effective dimension; otherwise, a price proportional to the minimal principal angle is paid. With the above results in hand, we can now control the excess risk of the ParK estimator in terms of the global norm of the target function and the global effective dimension. This allows to compare the performance of our partitioned method to that of a typical global method.

**Theorem 4.** *Let $\delta \in (0,1)$. Under the same assumptions of Proposition 1, let $\overline{q} = \mathrm{argmin}_{q \in [Q]}\rho_q$, for $0 < \lambda \leq \rho_{\overline{q}}\kappa^2$ when $\lambda_q = \lambda\rho_q^{-1}$ for each $q \in [Q]$, with probability at least $1 - 4\delta$,*

$$\mathcal{R}(\overline{f}_t) \leq 16(1 + Q^2\cos(\theta))\|f_*\|_{\mathcal{H}}^2\lambda + \frac{4\sigma^2}{n}\left(1 + \kappa^2\frac{\cos^2(\theta)}{\lambda}\right)\mathcal{N}(\lambda)\log(\frac{1}{\delta}).$$

If we consider a model where $\mathcal{H}$ is the orthogonal sum of the subspaces $\mathcal{H}_q$, as in [24], then $\cos(\theta) = 0$, and the bound of Theorem 4 simplifies to $\mathcal{R}(\overline{f}_t) = O(\lambda\|f_*\|^2 + \mathcal{N}(\lambda)n^{-1})$. In particular, setting $\lambda = O(1/\sqrt{n})$, we obtain the learning rate $O(1/\sqrt{n})$. This is known to be the optimal rate, in the minimax sense, for global KRR models [7]. Note that, in the orthogonal case, the constraint $\lambda \lesssim \rho_q$ translates to the minimal local point requirement $n_q \gtrsim \sqrt{n}$ for all $q$, and hence to a bound on the partition size, namely $Q \lesssim \sqrt{n}$. On the other hand, when the subspaces $\mathcal{H}_q$ are not perfectly orthogonal, our bound manifests a statistical-computational tradeoff, which is however quantified by the minimal principal angle. Further, the constraints on the local number of Nyström centers $m_q$, iterations $t_q$ of Proposition 1, and the choice $\lambda_q = \lambda\rho^{-1}$ with $\lambda = 1/\sqrt{n}$ to achieve the minimax rate, allows to recover a time complexity of $O\left(Q^2 n\log(n) + \sum_{q\in[Q]} n_q\sqrt{\frac{n_q}{\sqrt{n}}}\log(\frac{n_q}{\sqrt{n}})\right)$.

Analyses of (input space) partitioned kernel estimators have been conducted within different models, such as Gaussian SVM's on Voronoi partitions [23], general kernels on clusters [34], and block-diagonal kernels on arbitrary partitions [24]. In these works, the bounds are established in random design, for plain [23, 34] or Nyström [24] local KRR estimators. Our result is in fixed design, but compared to [24] incorporates the additional algorithmic ingredient of iterative optimization. For a perfectly orthogonal model ($\cos\theta = 0$), we recover the result in [24] as a special case (although in fixed design). In [34], the bias is controlled choosing same $\lambda_q$ for all $q$, while the crucial bound of Proposition 3 is made as an assumption. Note however that, at least in our proof of Proposition 3, it is important to choose a differently scaled $\lambda_q$ for each cell. Furthermore, our analysis and numerical tests motivate that partitioning the feature space is key to control both local bias and local effective dimension. Rather than on computational aspects, [23] focuses on extending statistical optimality for functions of local smoothness. This theme is also explored in [34, 24]. However, since the proposed partitioning step is either unsupervised [23, 34] or unspecified [24], improved rates can be obtained only under oracle assumptions, that is, assuming that the smoothness of the target function is localized right on the cells of the chosen partition. Partitions adapting to the unknown local smoothness of the target function can arguably be learned only in a supervised manner. This has been done for piecewise polynomial regression drawing ideas from multi-resolution analysis [4, 18]. An application of these ideas for kernel methods is not straightforward due to the usual computational constraints, but could be subject of future work.

## 5 Experiments

In this section we study the performance of ParK on some large-scale datasets ($n \approx 10^6, 10^7, 10^9$). In particular we consider dataset where at the moment, because of their cardinality, only a few solvers can efficiently learn from. For this reason we compare to the global large-scale kernel method FALKON which has so far being the method that performs the best in terms of time and accuracy on these datasets [22]. A standard divide-and-conquer method can not run on these datasets (for the high space complexity), for this reason we compare with a version where each local estimator is a sketched KRR estimator computed with FALKON. We run two different versions of this algorithm, D&C-FALK(v1) and D&C-FALK(v2), that differ only in their hyper-parameters choices as specified later in this section. We also consider a second version of ParK where the centroids of the partition's cells are chosen as $\{\phi(c_q)\}_{q=1}^Q$ with $c_q$ selected uniformly at randomly from the training data $X$ (referred to as ParK-Uni). For each experiment we report mean and standard deviation on 10 trials. The experiments are implemented in python using pytorch and the FALKON library [22]. The experiments run on a machine with 2 Intel Xeon Silver 4116 CPUs and 1 GPU NVIDIA Titan Xp. The ram of the machine is 256 GB. We perform experiments on the four large-scale datasets TAXI ($n \approx 10^9$, $d = 9$, regression), HIGGS ($n \approx 10^7$, $d = 28$, classification), AIRLINE ($n \approx 10^6$, $d = 8$, regression), AIRLINE-CLS ($n \approx 10^6$, $d = 8$, classification) with the same pre-processing and same random train/test split used in [22]. We do not cross validate hyper-parameters of the local estimators of ParK. Instead we use the same used by FALKON in the paper [22] with the following exeptions: let $\lambda$ be the global regularization parameters of FALKON and $m$ the number of the Nyström points, the local estimators of ParK use regularization $\lambda_q = \lambda\rho_q^{-1}$ and $m_q = m\rho_q$ as suggested by the theory. D&C-FALK(v1) also follows the same rule for setting the hyper-parameters of its local estimators, while D&C-FALK(v2) uses the same of the (v1) version except the number of Nyström centers which are $3m_q$ in AIRLINE and AIRLINE-CLS, $5m_q$ in HIGGS, and $6m_q$ in TAXI. The number of centroids used by ParK and D&C-FALK is $Q = 32$ for all experiments. Performance for different $Q$ values remains almost identical but worsen in time for higher values. Further, note that the local estimators of ParK and D&C-FALK are learned sequentially. We report in Table 2 the errors and times. In particular, for ParK(-Uni) we report the initialization time that include the greedy algorithm to select the centroids (not for ParK-Uni) and the assignment of the training points to the corresponding cell, the sequential training times of the local estimators, and the total time of this pipeline.

We can see that ParK can match the accuracy of the global FALKON estimator with a smaller computational cost. ParK-Uni requires further less time, at the expense of some loss in accuracy, confirming that a worse partition can affect generalization, as suggested by our theory. The reason of the ParK-Uni speedup is twofold. First, the initialization step requires only to assign points to a set of randomly selected centroids, and second, the local subsets of points have uniform cardinality (which is usually not the case for normal ParK). D&C-FALK(v1) is the algorithm with with smallest

Table 2: Accuracy and running time comparison on large-scale datasets.

| | TAXI $n \approx 10^9$ | | | | HIGGS $n \approx 10^7$ | | | |
| --- | --- | --- | --- | --- | --- | --- | --- | --- |
| | ERROR (RMSE) | TIME (MIN.) | | | ERROR (1−AUC) | TIME (SEC.) | | |
| | | INIT | TRAIN | TOTAL | | INIT | TRAIN | TOTAL |
| ParK | 312.0±0.2 | 25±1 | 39±13 | 64±13 | 0.182±0.001 | 30±2 | 474±172 | 504±172 |
| ParK-Uni | 315.7±0.6 | 5±1 | 13±1 | 18±1 | 0.192±0.000 | 3±1 | 67±7 | 70±7 |
| Falkon | 311.7±0.1 | - | - | 120±1 | 0.180±0.001 | - | - | 715±6 |
| D&C-Falk(v1) | 356.2±0.2 | - | - | 14±1 | 0.212±0.000 | - | - | 50±1 |
| D&C-Falk(v2) | 327.4±0.1 | - | - | 29±1 | 0.195±0.000 | - | - | 288±2 |

| | AIRLINE $n \approx 10^6$ | | | | AIRLINE-CLS $n \approx 10^6$ | | | |
| --- | --- | --- | --- | --- | --- | --- | --- | --- |
| | ERROR (MSE) | TIME (SEC.) | | | ERROR (C-ERR) | TIME (SEC.) | | |
| | | INIT | TRAIN | TOTAL | | INIT | TRAIN | TOTAL |
| ParK | 0.760±0.005 | 6±1 | 71±9 | 77±10 | 31.5±0.2 % | 9±1 | 55±6 | 64±6 |
| ParK-Uni | 0.766±0.006 | 1±1 | 32±3 | 33±3 | 31.6±0.2 % | 2±1 | 22±2 | 24±2 |
| Falkon | 0.758±0.005 | - | - | 334±2 | 31.5±0.2 % | - | - | 391±5 |
| D&C-Falk(v1) | 0.834±0.005 | - | - | 27±1 | 33.2±0.1 % | - | - | 20±1 |
| D&C-Falk(v2) | 0.799±0.005 | - | - | 96±1 | 32.2±0.1 % | - | - | 73±1 |

training time but achieve significantly worse performance using the same rule to choose the number of Nyström points of ParK. For this reason, in D&C-FALK(v2) we increase the number of centroid to improve the performance, but the error of the method still results higher than the others with a training time now higher than ParK.

## 6 Conclusions and limitations.

In this paper we have proposed a new algorithm for large scale kernel ridge regression. Our method integrates and jointly exploits three previously uncombined algorithmic strategies, namely partitions, sketching and (preconditioned) iterative optimization. Distinctively from traditional partitioned methods, we have introduced the idea of partitioning the feature space, which allows to directly control and resolve the localization of the kernel model. We have presented a simple analysis that characterizes the statistical-computational trade-off of a partitioned kernel estimator by the interplay of intuitive quantities. Moreover, we have demonstrated that our algorithm performs favourably against a state-of-the-art large scale global method.

The main theoretical limitation of our work is the lack of a result connecting the proposed partitioning algorithm to the properties of the resulting partition. This seems to be a common gap in the literature of partitioned kernel methods, where partitions are often assumed to be given or, if explicitly constructed, are not statistically characterized. While our construction is theoretically motivated by the analysis and practically validated by the experiments, an actual guarantee is missing. In particular, one could try to prove that the proposed greedy procedure would actually find a maximally orthogonal decomposition of the hypothesis space, under suitable assumptions. From an algorithmic point of view, we point out that the computational cost of the greedy algorithm limits the choice of the partition size. Indeed, large partitions accelerate the training step, but increase the initialization time. We remark, however, that our model is flexible enough to include cheaper partitioning options. For example, our experiments show that uniformly chosen partitions can still produce good results.

### Acknowledgments and Disclosure of Funding

The authors thank Nicole Mücke for her useful feedback. This material is based upon work supported by the Center for Brains, Minds and Machines (CBMM), funded by NSF STC award CCF-1231216, and the Italian Institute of Technology. We gratefully acknowledge the support of NVIDIA Corporation for the donation of the Titan Xp GPUs and the Tesla k40 GPU used for this research. L. R. acknowledges the financial support of the European Research Council (grant SLING 819789), the AFOSR projects FA9550-18-1-7009, FA9550-17-1-0390 and BAA-AFRL-AFOSR-2016-0007 (European Office of Aerospace Research and Development), and the EU H2020-MSCA-RISE project NoMADS - DLV-777826.

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
