# A    Appendix

## A.1    Relevant operators

We define operators in global, local and subsampled variants. The global definitions are standard in the analysis of kernel methods [7]. In view of (3), we assume without loss of generality that $\mathcal{H} = \mathcal{H}_n$. Recall that $L^2 = L^2(\rho)$ with $\rho = \frac{1}{n}\sum_{i=1}^{n}\delta_{x_i}$, and that we identify $L^2$ with $\mathbb{R}^n$ with inner product $\langle u, w \rangle_{L^2} = \frac{1}{n}u^\top w$.

**Global operators:**

- $S : \mathcal{H} \to L^2$     $Sf(x) = \langle f, \phi(x) \rangle_{\mathcal{H}}$     the sampling operator
- $S^* : L^2 \to \mathcal{H}$     $S^*w = \frac{1}{n}\sum_{i\in[n]}w_i\,\phi(x_i)$     the out-of-sample extension operator
- $T : \mathcal{H} \to \mathcal{H}$     $T = S^*S = \frac{1}{n}\sum_{i\in[n]}\phi(x_i)\otimes\phi(x_i)$     the covariance operator

We now define local versions of the operators above, conditioned on the partitioning (4). Thanks to (10), we can assume without loss of generality that $\mathcal{H}_q = \mathrm{span}\{\phi(x_i) : i \in [n]_q\}$. Let $L_q^2 = L^2(\rho(\cdot \mid V_q))$ with $\rho(\cdot \mid V_q) = \frac{1}{n_q}\sum_{i\in[n]_q}\delta_{x_i}$. We identify $L_q^2$ with $\mathbb{R}^{n_q}$ endowed with the inner product $\langle u, w \rangle_{L_q^2} = \frac{1}{n_q}u^\top w$.

**Local operators:**

- $S_q : \mathcal{H} \to L_q^2$     $S_q f(x) = \langle f, \phi(x) \rangle_{\mathcal{H}}$
- $S_q^* : L_q^2 \to \mathcal{H}$     $S_q^* w = \frac{1}{n_q}\sum_{i\in[n]_q}w_i\,\phi(x_i)$
- $T_q : \mathcal{H} \to \mathcal{H}$     $T_q = S_q^* S_q = \frac{1}{n_q}\sum_{i\in[n]_q}\phi(x_i)\otimes\phi(x_i)$

The orthogonal projection $P_q : \mathcal{H} \to \mathcal{H}$ onto the subspace $\mathcal{H}_q$ is given by

$$P_q = S_q^+ S_q,$$

where $^+$ denotes the Moore–Penrose pseudoinverse. Let $\rho_q = \rho(V_q) = n_q/n$. We observe that

$$T = \sum_q T_q \rho_q, \tag{16}$$

namely, the global covariance is an average of local covariances. Based on the local subsampling (11), we further introduce the following operators.

**Subsampled local operators:**

- $\widetilde{S}_q : \mathcal{H} \to \mathbb{R}^{m_q}$     $\widetilde{S}_q f = \frac{1}{\sqrt{m_q}}(\langle f, \phi(\widetilde{x}_{q,i})\rangle)_{i=1}^{m_q}$
- $\widetilde{S}_q^* : \mathbb{R}^{m_q} \to \mathcal{H}$     $\widetilde{S}_q^* w = \frac{1}{\sqrt{m_q}}\sum_{i=1}^{m_q}w_i\phi(\widetilde{x}_{q,i})$

## A.2    Controlling the excess risk

In this section we prove Proposition 1. Both the Euclidean norm of vectors and the spectral norm of matrices are denoted by $\|\cdot\|$.

**From global to local excess risk.**    Note that another way to write the excess risk (15) is

$$\mathcal{R}(\widehat{f}) = \|T^{1/2}(\widehat{f} - f_*)\|_{\mathcal{H}}^2.$$

Define now a local version of the above risk on the cells of the partition (4) as

$$\mathcal{R}_q(\widehat{f}_q) = \|S_q(\widehat{f}_q - f_*)\|_{L_q^2}^2. \tag{17}$$

**Lemma 1.** *For every $\overline{f}$ defined as in (9),*

$$\mathcal{R}(\overline{f}) = \sum_{q=1}^{Q}\mathcal{R}_q(\widehat{f}_q)\rho_q.$$

*Proof.* We have

$$
\begin{aligned}
\mathcal{R}(\overline{f}) &= \|\overline{f} - f_*\|_{L^2}^2 \\
&= \sum_{q \in [Q]} \sum_{i \in [n]_q} |\overline{f}(x_i) - f_*(x_i)|^2 \\
&= \sum_{q \in [Q]} \sum_{i \in [n]_q} |\widehat{f}_q(x_i) - f_*(x_i)|^2 \\
&= \sum_{q \in [Q]} \sum_{i \in [n]_q} |S_q \widehat{f}_q(x_i) - S_q f_*(x_i)|^2 \\
&= \sum_{q \in [Q]} \|S_q(\widehat{f}_q - f_*)\|_{L_q^2}^2 \, \rho_q.
\end{aligned}
$$

$\square$

**Lemma 2.** *The local excess risk* (17) *can be rewritten as*

$$
\mathcal{R}_q(\widehat{f}_q) = \|T_q^{1/2}(\widehat{f}_q - P_q f_*)\|_{\mathcal{H}}^2.
$$

*Proof.* Since $S_q = S_q S_q^+ S_q = S_q P_q$, we have

$$
\begin{aligned}
\|S_q(\widehat{f}_q - f_*)\|_{L_q^2}^2 &= \|S_q(\widehat{f}_q - P_q f_*)\|_{L_q^2}^2 \\
&= \langle S_q(\widehat{f}_q - P_q f_*), S_q(\widehat{f}_q - P_q f_*) \rangle_{L_q^2} \\
&= \langle T_q(\widehat{f}_q - P_q f_*), (\widehat{f}_q - P_q f_*) \rangle_{\mathcal{H}} \\
&= \|T_q^{1/2}(\widetilde{f}_{q,t} - P_q f_*)\|_{\mathcal{H}}^2.
\end{aligned}
$$

$\square$

**From FALKON to Nyström local estimators.** We now control the local excess risk of each local estimator $\widetilde{f}_{q,t}$ as defined in (13) with the exact local Nyström estimator defined by

$$
\widetilde{f}_q(x) = \sum_{i=1}^{m_q} (\widetilde{\alpha}_q)_i K(\widetilde{x}_i, x), \qquad \widetilde{\alpha}_q = \underset{\alpha \in \mathbb{R}^{m_q}}{\operatorname{argmin}} \frac{1}{n_q} \|K_{n_q m_q} \alpha - Y_q\|^2 + \lambda_q \alpha^\top K_{m_q} \alpha. \tag{18}
$$

Adapting the analysis of [29] to fixed design and local setting we derive the following lemma.

**Lemma 3.** *Let $\delta \in (0, 1]$, the Nyström centers in $\widetilde{f}_{q,t}$ be selected uniformly at random from $X_q$, $n_q, m_q, t \in \mathbb{N}$. If $0 \leq \lambda_q \leq \kappa^2$ and*

$$
m_q \geq 5[1 + 14\mathcal{N}_{\infty,q}(\lambda_q)] \log(\frac{8\kappa^2}{\lambda_q \delta}), \tag{19}
$$

*then, with probability $1 - 2\delta$,*

$$
\mathcal{R}_q(\widetilde{f}_{q,t})^{1/2} \leq \mathcal{R}_q(\widetilde{f}_q)^{1/2} + 6\sigma\kappa \|P_q f_*\|_{\mathcal{H}} \log(\frac{1}{\delta}) e^{-\frac{t}{2}}.
$$

*Proof.* We follow the proof of Theorem 1 and Lemma 11 of [29], replacing the operators $S, S^*, C$ in [29] with our local operators $S_q, S_q^*, T_q$. Note that in fixed design we do not have population operators, hence we can upper bound deterministically quantities that in random design require concentration arguments. Moreover, we upper bound the quantity $\frac{\|Y_q\|}{\sqrt{n_q}}$ (our equivalent of $\widehat{\nu}$ in Theorem 1 of [29]) as follows. Recalling (1), we have

$$
\frac{\|Y_q\|}{\sqrt{n_q}} = \frac{1}{\sqrt{n_q}} \sqrt{\sum_{i \in [n]_q} (f_*(x_i) + \varepsilon_i)^2} \leq \frac{1}{\sqrt{n_q}} \left( \sqrt{2 \sum_{i \in [n]_q} f_*(x_i)^2} + \sqrt{2 \sum_{i \in [n]_q} \varepsilon_i^2} \right). \tag{20}
$$

Exploiting Assumption 1, for every $x \in X_q$ we have $f_*(x) = \langle f_*, K_x \rangle = \langle P_q f_*, K_x \rangle$. Thus, Assumption 2 gives

$$
\sup_{x \in X_q} |\langle P_q f_*, K_x \rangle_{\mathcal{H}}| \leq \sup_{x \in \mathcal{X}} \|P_q f_*\|_{\mathcal{H}} \|K_x\|_{\mathcal{H}} \leq \kappa \|P_q f_*\|_{\mathcal{H}}. \tag{21}
$$

Let $\widehat{\varepsilon} = [\varepsilon_i]_{i \in [n]_q} \in \mathbb{R}^{n_q}$, Then, by Assumption 3, using Lemma 19 of [17] we obtain that, with probability at least $1 - \delta$,

$$\|\widehat{\varepsilon}\| \leq \sigma \sqrt{n_q} \log(\frac{1}{\delta}). \tag{22}$$

Plugging (21) and (22) in (20) we conclude the proof. $\qquad\square$

We now control the local excess risk of each local exact Nyström estimator $\widetilde{f}_q$ as defined in (18). Adapting the analysis of [28] locally to fixed design we derive the following lemma.

**Lemma 4.** *Let $\delta \in (0,1]$, the Nyström centers in $\widetilde{f}_q$ be selected uniformly at random from $X_q$, $n_q, m_q, t \in \mathbb{N}$. If $0 \leq \lambda_q \leq \kappa^2$ and*

$$m_q \geq [2 + 3\mathcal{N}_{\infty,q}(\lambda_q)] \log(\frac{8\kappa^2}{\lambda_q \delta}),$$

*then, with probability $1 - 2\delta$,*

$$\mathcal{R}_q(\widetilde{f}_q)^{1/2} \leq 3 \|P_q f_*\|_{\mathcal{H}} \sqrt{\lambda_q} + \frac{\sigma}{\sqrt{n_q}} \sqrt{\mathcal{N}_q(\lambda_q) + \sqrt{\mathcal{N}_q(\lambda_q) \log(\frac{1}{\delta})} + 2\log(\frac{1}{\delta})}.$$

*Proof.* We follow the proof of Theorem 2 and Proposition 2 of [28], replacing the operators $S, S^*, C$ in [28] with our local operators $S_q, S_q^*, T_q$. As for the proof of Lemma 3, concentration inequalities for empirical operators are replaced by deterministic bounds. Further we need to control the sample error in Lemma 4 of [28] with a different concentration argument. Let $\widehat{Y}_q = \frac{\|Y_q\|}{\sqrt{n_q}}$. The sample error in fixed design is

$$\left\| (T_q + \lambda_q)^{1/2} S_q^* (\widehat{Y}_q - S_q P_q f_*) \right\|_{\mathcal{H}}.$$

In view of (1) we have

$$\left\| (T_q + \lambda_q)^{1/2} S_q^* (\widehat{Y}_q - S_q P_q f_*) \right\|_{\mathcal{H}} = \frac{1}{\sqrt{n_q}} \left\| (T_q + \lambda_q)^{1/2} S_q^*(\widehat{\varepsilon}) \right\|_{\mathcal{H}},$$

where $\widehat{\varepsilon} = [\varepsilon_i]_{i \in [n]_q} \in \mathbb{R}^{n_q}$. Now using Assumption 3, Remark 2.2 of [16] and the definition of local effective dimension, we obtain, with probability at least $1 - \delta$,

$$\frac{1}{\sqrt{n_q}} \left\| (T_q + \lambda_q)^{1/2} S_q^*(\widehat{\varepsilon}) \right\|_{\mathcal{H}} \leq \frac{\sigma}{\sqrt{n_q}} \sqrt{\mathcal{N}_q(\lambda_q) + \sqrt{\mathcal{N}_q(\lambda_q) \log(\frac{1}{\delta})} + 2\log(\frac{1}{\delta})},$$

which concludes the proof. $\qquad\square$

We are now ready to prove Proposition 1.

**Proof of Proposition 1** From Lemmas 3 and 4 we know that, under their respective assumptions and for a value of $m_q$ as in (19), with probability $1 - 4\delta$,

$$\mathcal{R}_q(\widetilde{f}_{q,t})^{1/2} \leq 3 \|P_q f_*\|_{\mathcal{H}} \sqrt{\lambda_q} + \frac{\sigma}{\sqrt{n_q}} \sqrt{\mathcal{N}_q(\lambda_q) + \sqrt{\mathcal{N}_q(\lambda_q) \log(\frac{1}{\delta})} + 2\log(\frac{1}{\delta})}$$

$$+ 6\sigma\kappa \|P_q f_*\|_{\mathcal{H}} \log(\frac{1}{\delta}) e^{-\frac{t}{2}}.$$

We consider now a number of iterations $t$ such that $6\sigma\kappa \|P_q f_*\|_{\mathcal{H}} \log(\frac{1}{\delta}) e^{-\frac{t}{2}} \leq \|P_q f_*\|_{\mathcal{H}} \sqrt{\lambda_q}$, that is

$$t \geq 2 \log\left( \frac{6\sigma\kappa \log(1/\delta)}{\sqrt{\lambda_q}} \right).$$

Under the above constraint on $t$ we can rewrite the upper bound on the risk

$$\mathcal{R}_q(\widetilde{f}_{q,t})^{1/2} \leq 4 \|P_q f_*\|_{\mathcal{H}} \sqrt{\lambda_q} + \frac{\sigma}{\sqrt{n_q}} \sqrt{\mathcal{N}_q(\lambda_q) + \sqrt{\mathcal{N}_q(\lambda_q) \log(\frac{1}{\delta})} + 2\log(\frac{1}{\delta})}.$$

We can now collect the local excess risk bounds above for all $q \in [Q]$ using Lemmas 1 and 2, concluding the proof. $\qquad\square$

### A.3 Controlling the partition

In this section we prove Propositions 2 and 3. With a slight abuse of notation, the operator norm on $\mathcal{H}$ is denoted by $\| \cdot \|_{\mathcal{H}}$.

*Proof of Proposition 2.* We have

$$\sum_q \|P_q f_*\|_{\mathcal{H}}^2 = \sum_q \langle P_q f_*, P_q f_* \rangle_{\mathcal{H}} = \sum_q \langle f_*, P_q f_* \rangle_{\mathcal{H}} = \langle f_*, \sum_q P_q f_* \rangle_{\mathcal{H}} \le \| \sum_q P_q \|_{\mathcal{H}} \|f_*\|_{\mathcal{H}}^2.$$

Now, let $U_q : \mathcal{H} \to \mathbb{R}^{n_q}$ such that $U_q^* U_q = P_q$, $U_q U_q^* = I_{n_q}$, and define

$$U = [U_1, \ldots, U_Q]^\top : \mathcal{H} \to \mathbb{R}^n.$$

Then $\sum_q P_q = U^* U$, and

$$\| \sum_q P_q \|_{\mathcal{H}} = \|U^* U\|_{\mathcal{H}} = \|UU^*\|,$$

Let $W = UU^*$. Then $W \in \mathbb{R}^{n \times n}$ is built as

$$W = \begin{bmatrix} U_1 U_1^* & \cdots & U_1 U_Q^* \\ \vdots & \ddots & \vdots \\ U_Q U_1^* & \cdots & U_Q U_Q^* \end{bmatrix} = \begin{bmatrix} I_{n_1} & \cdots & W_{1,Q} \\ \vdots & \ddots & \vdots \\ W_{Q,1} & \cdots & I_{n_Q} \end{bmatrix}.$$

Thus, for $a = [a_1, \ldots, a_Q] \in \mathbb{R}^n$, $a_q \in \mathbb{R}^{n_q}$, we have

$$\begin{aligned}
\|UU^*\| &= \|W\| \\
&= \|I + (W - I)\| \\
&= 1 + \lambda_{\max}(W - I) \\
&= 1 + \max_{\|a\|=1} a^\top (W - I) a \\
&= 1 + \max_{\|a\|=1} \sum_q a_q^\top (W_{q,q} - I_{n_q}) a_q + \sum_{q,k : q \ne k} a_q^\top (W_{q,k} - 0) a_k \\
&= 1 + \max_{\|a\|=1} \sum_q a_q^\top 0 a_q + \sum_{q,k : q \ne k} a_q^\top W_{q,k} a_k \\
&= 1 + \max_{\|a\|=1} \sum_{q,k : q \ne k} a_q^\top W_{q,k} a_k \\
&\le 1 + \sum_{q,k : q \ne k} \max_{\|a\|=1} a_q^\top W_{q,k} a_k.
\end{aligned}$$

Now we can bound

$$\sum_{q,k : q \ne k} \max_{\|a\|=1} a_q^\top W_{q,k} a_k \le Q^2 \max_{q,k : q \ne k} \max_{\|a\|=1} a_q^\top W_{q,k} a_k,$$

and reparameterizing $a_q = \beta_q b_q$, $\beta_q \ge 0$, $b_q \in \mathbb{R}^{n_q}$, we get

$$\begin{aligned}
\max_{\|a\|=1} a_q^\top W_{q,k} a_k &= \max_{\substack{\|b_1\|=\cdots=\|b_Q\|=1 \\ \beta_1^2+\cdots+\beta_Q^2=1}} \beta_q \beta_k b_q^\top W_{q,k} b_k \\
&= \max_{\beta_1^2+\cdots+\beta_Q^2=1} \beta_q \beta_k \max_{\|b_1\|=\cdots=\|b_Q\|=1} b_q^\top W_{q,k} b_k \\
&= \max_{\beta_1^2+\cdots+\beta_Q^2=1} \beta_q \beta_k \max_{\|b_q\|=\|b_k\|=1} b_q^\top W_{q,k} b_k \\
&= \max_{\beta_1^2+\cdots+\beta_Q^2=1} \beta_q \beta_k \cos(\angle(\mathcal{H}_q, \mathcal{H}_k)) \\
&\le \cos(\theta).
\end{aligned}$$

Putting all together, we finally obtain

$$\| \sum_q P_q \|_{\mathcal{H}} \le 1 + Q^2 \cos(\theta),$$

which completes the proof. $\qquad\qquad\qquad\qquad\qquad\qquad\qquad\qquad\qquad\qquad\qquad\qquad\qquad\square$

*Proof of Proposition 3.* Let $\widetilde{T}_q = P_q T P_q \rho_q^{-1}$, and let $M_q = (\widetilde{T}_q + \lambda_q)^{1/2}(T_q + \lambda_q)^{-1}(\widetilde{T}_q + \lambda_q)^{1/2}$.
Then

$$
\sum_q \mathcal{N}_q(\lambda_q) = \sum_q \mathrm{Tr}(S_q(T_q + \lambda_q)^{-1}S_q^*)
$$

$$
= \sum_q \mathrm{Tr}(S_q(\widetilde{T}_q + \lambda_q)^{-1/2}M_q(\widetilde{T}_q + \lambda_q)^{-1/2}S_q^*)
$$

$$
= \sum_q \mathrm{Tr}(M_q(\widetilde{T}_q + \lambda_q)^{-1/2}S_q^*S_q(\widetilde{T}_q + \lambda_q)^{-1/2})
$$

$$
\leq \sup_q \|M_q\|_{\mathcal{H}} \sum_q \mathrm{Tr}((\widetilde{T}_q + \lambda_q)^{-1/2}S_q^*S_q(\widetilde{T}_q + \lambda_q)^{-1/2})
$$

$$
= \sup_q \|M_q\|_{\mathcal{H}} \sum_q \mathrm{Tr}(S_q(\widetilde{T}_q + \lambda_q)^{-1}S_q^*),
$$

where in the third and last equalities we used the cyclic property of the trace, and in the fourth step
we applied Holder's inequality. We first bound the trace. We have

$$
(\widetilde{T}_q + \lambda_q)^{-1} = \lambda_q^{-1}(\widetilde{T}_q + \lambda_q - \widetilde{T}_q)(\widetilde{T}_q + \lambda_q)^{-1}
$$

$$
= \lambda_q^{-1}(I - \widetilde{T}_q(\widetilde{T}_q + \lambda_q)^{-1})
$$

$$
= \lambda_q^{-1}(I - P_q S^* S P_q \rho_q^{-1}(P_q S^* S P_q \rho_q^{-1} + \lambda_q)^{-1})
$$

$$
= \lambda_q^{-1}(I - P_q S^* S P_q(P_q S^* S P_q + \lambda_q \rho_q)^{-1})
$$

$$
= \lambda_q^{-1}(I - P_q S^*(S P_q S^* + \lambda_q \rho_q)^{-1} S P_q)
$$

$$
\preceq \lambda_q^{-1}(I - P_q S^*(S S^* + \lambda_q \rho_q)^{-1} S P_q).
$$

where the fifth equality follows from the Woodbury identity. Thus, multiplying by $S_q$ from the left
and by $S_q^*$ from the right, we get

$$
S_q(\widetilde{T}_q + \lambda_q)^{-1}S_q^* \preceq \lambda_q^{-1}(S_q S_q^* - S_q P_q S^*(S S^* + \lambda_q \rho_q)^{-1} S P_q S_q^*)
$$

$$
= \lambda_q^{-1}(S_q S_q^* - S_q S^*(S S^* + \lambda_q \rho_q)^{-1} S S_q^*)
$$

$$
= \lambda_q^{-1}(S_q(I - S^*(S S^* + \lambda_q \rho_q)^{-1} S)S_q^*)
$$

$$
= \lambda_q^{-1}(S_q(I - (T + \lambda_q \rho_q)^{-1} T)S_q^*)
$$

$$
= \lambda_q^{-1}(S_q(T + \lambda_q \rho_q)^{-1}(T + \lambda_q \rho_q - T)S_q^*)
$$

$$
= S_q(T + \lambda_q \rho_q)^{-1}S_q^* \rho_q,
$$

where again the fourth equality follows from the Woodbury identity. Therefore,

$$
\mathrm{Tr}(S_q(\widetilde{T}_q + \lambda_q)^{-1}S_q^*) \leq \mathrm{Tr}(S_q(T + \lambda_q \rho_q)^{-1}S_q^* \rho_q) = \mathrm{Tr}((T + \lambda_q \rho_q)^{-1}T_q \rho_q).
$$

Setting $\lambda_q = \lambda \rho_q^{-1}$ and using (16) we obtain

$$
\sum_q \mathrm{Tr}(S_q(\widetilde{T}_q + \lambda_q)^{-1}S_q^*) \leq \mathrm{Tr}((T + \lambda)^{-1} \sum_q T_q \rho_q) = \mathrm{Tr}((T + \lambda)^{-1}T) = \mathcal{N}(\lambda).
$$

We next bound $\|M_q\|_{\mathcal{H}}$. The operators $T_q + \lambda_q$ and $\widetilde{T}_q + \lambda_q$ are invertible, hence $M_q$ shares the same
spectrum as $(T_q + \lambda_q)^{-1/2}(\widetilde{T}_q + \lambda_q)(T_q + \lambda_q)^{-1/2}$, and in particular

$$
\|M_q\|_{\mathcal{H}} = \|(T_q + \lambda_q)^{-1/2}(\widetilde{T}_q + \lambda_q)(T_q + \lambda_q)^{-1/2}\|_{\mathcal{H}}.
$$

Now, let

$$
\overline{T}_q : \mathcal{H} \to \mathcal{H} \qquad \overline{T}_q = \frac{1}{n} \sum_{\phi(x_i) \notin V_q} \phi(x_i) \otimes \phi(x_i).
$$

Then $T = T_q \rho_q + \overline{T}_q$, and

$$
\widetilde{T}_q = P_q T_q P_q + P_q \overline{T}_q P_q \rho_q^{-1} = T_q + P_q \overline{T}_q P_q \rho_q^{-1}.
$$

Therefore,

$$
\begin{aligned}
\|M_q\|_{\mathcal{H}} &= \|(T_q + \lambda_q)^{-1/2}(T_q + \lambda_q + P_q\overline{T}_q P_q \rho_q^{-1})(T_q + \lambda_q)^{-1/2}\|_{\mathcal{H}} \\
&= \|I + \rho_q^{-1}(T_q + \lambda_q)^{-1/2}(P_q\overline{T}_q P_q)(T_q + \lambda_q)^{-1/2}\|_{\mathcal{H}} \\
&\leq 1 + \rho_q^{-1}\|T_q + \lambda_q)^{-1/2}\|_{\mathcal{H}}\|P_q\overline{T}_q P_q\|_{\mathcal{H}}\|T_q + \lambda_q)^{-1/2}\|_{\mathcal{H}} \\
&\leq 1 + \rho_q^{-1}\lambda_q^{-1/2}\|P_q\overline{T}_q P_q\|_{\mathcal{H}}\lambda_q^{-1/2} \\
&= 1 + \tfrac{1}{\lambda_q \rho_q}\|P_q\overline{T}_q P_q\|_{\mathcal{H}}.
\end{aligned}
$$

For $\lambda_q = \lambda \rho_q^{-1}$, we get

$$
\|M_q\|_{\mathcal{H}} \leq 1 + \tfrac{1}{\lambda}\|P_q\overline{T}_q P_q\|_{\mathcal{H}}.
$$

Finally,

$$
\begin{aligned}
\|P_q\overline{T}_q P_q\|_{\mathcal{H}} &\leq \frac{1}{n}\sum_{\phi(x_i)\notin V_q}\|P_q\phi(x_i)\|_{\mathcal{H}}^2 \\
&= \frac{1}{n}\sum_{\phi(x_i)\notin V_q}\|\phi(x_i)\|_{\mathcal{H}}^2\cos^2(\angle(\phi(x_i),\mathcal{H}_q)) \\
&\leq \frac{1}{n}\sum_{\phi(x_i)\notin V_q}\sup_i\|\phi(x_i)\|_{\mathcal{H}}^2\cos^2(\min_{k\neq q}\angle(V_q,\mathcal{H}_q)) \\
&= \sup_i\|\phi(x_i)\|_{\mathcal{H}}^2\cos^2(\min_{k\neq q}\angle(V_q,\mathcal{H}_q))\frac{1}{n}(n-n_q) \\
&\leq \sup_i\|\phi(x_i)\|_{\mathcal{H}}^2\cos^2(\min_{k\neq q}\angle(V_q,\mathcal{H}_q)),
\end{aligned}
$$

which leads to the desired bound. $\qquad\square$