# OpenReview forum: "ParK: Sound and Efficient Kernel Ridge Regression by Feature Space Partitions"
_NeurIPS.cc/2021/Conference — NeurIPS 2021 Poster_

### Official Review · Reviewer_gQRE · 2021-07-01

**Rating:** 5
**Confidence:** 3

**Summary:**


This work introduces a new approach to large scale KRR, ParK.
This approach relies on the combination of two main elements :
a Voronoi partition of the feature space that aims at maximizing the first principal angle  of the partition,
and a careful use of the FALKON estimator.
The authors provide a theoretical upper bound of the risk of ParK, and compare the performance of ParK to FALKON on different experiments.


**Limitations And Societal Impact:**

See questions above.


**Main Review:**

While the use of the FALKON estimator is not new, the idea of splitting the feature space using a Voronoi partition is original (to the best of my knowledge), appears natural and yields interesting results.
On the one hand, I found the link between the first principal angle of the partition and the risk of the ParK estimator to be particularly insightful (Theorem 1, Prop 2 & 3).
On the other hand, the experimental evaluation seems a bit insufficient, and some statement could benefit from being clarified. These points are developed below.
If the authors can address these questions, I am willing to increase my  rating.

## Questions :


### Q1
The authors claim (line 161) that the complexity of finding the centroid is $O(Q^2n \log n)$.
However, given equation (6), shouldn't the complexity be $\sum_q O (n q^3) \approx O(nQ^4) $?
indeed, each centroid requires the inversion of $K_q$, a matrix of size $q \times q$.
This is particularly important in the case where $Q$ is comparable to $\sqrt{n}$, discussed by the authors line 252-260, as it drastically changes the complexity of ParK.


###  Q2
 I do not understand the inequality  line 208
$$  \mathcal{N}_{\infty,q} \le \lambda_q^{-1} \lambda_\max{}(T_q)  $$
Shouldn't  $\lambda_\min{}(T_q)$ appear instead ? Can the authors elaborate on the proof of this inequality ?

### Q3

While Assumption 1 is not unheard of in kernel theory, in this paper the authors also make the hypothesis
$H=H_n$ . Hence, in the paper, the resulting assumption is $f_* \in H_n,$  i.e. $ f_*$ can be perfectly estimated given the observations. This assumption seems too strong for me, can the authors discuss this question ?


### Q4. About experiments

While the results of the experimental evaluation look promising, the following elements appear to be missing.

1.Which kernel  did the authors chose? This is particularly key for ParK, as the kernel also influences the partition. Also, it would be interesting to study the influence of the choice of the kernel, e.g. by trying multiple kernel on each datasets.

2.In the experiments the authors limit themselves to $Q=32$.  As a consequence, $n_q$ is close to $n$, and the benefit of ParK on time complexity are more limited.  Can the authors elaborate on the consequences of $Q >> 1$ in practice ?

3.Surprisingly, the loss of performance resulting from using random partitioning (Uni-Park) instead of Voronoi partitioning (Park) is small. Can the authors elaborate on the apparent  small benefits produced by their partitioning  method ?

4.Finally, I think it would be particularly interesting to report the first principle angle of the partition in each dataset, and compare it to the results of ParK.

=========================================
Update after rebuttal.

Thank you for answering the questions, in particular for clarifying Q2. However, after carefully considering the authors' response, I still consider that the experimental evaluation of the method is lacking in its current form. Therefore I only change my evaluation from 4 to 5.  I suggest that the authors add all the new results they mentioned in their rebuttal to the paper for future submissions.


**Time Spent Reviewing:**

4

---

> ### Author Response · Authors · 2021-08-10
> **Reply to Reviewer gQRE**
>
> We thank the reviewer for their useful feedback. We answer in the following the questions and clarifications presented in the review, and we will add a paragraph in the main paper discussing each one of them
>
> 1) We simplify the complexity from $O(Q^2n + Q n \log(n))$ to $O(Q^2 n \log(n))$ for brevity. Further note that the computational complexity of the greedy algorithm is $O(Q^2 n)$ because the inversion of the $Q \times Q$ matrix can be efficiently computed using rank-1 updates.
>
> 2) We thank the reviewer for noticing that actually the inequality contains a typo: $ \lambda_\max $ should be $ \kappa^2 $. We report the proof below. This typo does not propagate in the rest of the paper.
>
> $
>     \mathcal{N}_{\infty,q} (\lambda_q)
> $
>
> $
> = \sup_{x \in X_q} \langle \phi(x) , (T_q +\lambda_q)^{-1} \phi(x) \rangle
> $
>
> $
>  \le \Vert (T_q + \lambda_q)^{-1} \Vert \sup_{x \in X_q} \Vert \phi(x) \Vert^2
> $
>
> $
>  \le \lambda_q^{-1} \sup_{x} \Vert \phi(x) \Vert^2
> $
>
> $
>  \le \lambda_q^{-1} \kappa^2
> $
>
> 3) The fact that one may assume $f_* \in H_n$ is a common technicality resulting from the fixed design setting, where one is only concerned about in-sample estimation (see Bach, “Sharp analysis of low-rank kernel matrix approximations”). Roughly speaking, if $f_* \in H$, we can always replace $f_*$ with its projection on $H_n$, which might differ from $f_*$ out of the samples, but coincides with $f_*$ at the samples. If the reviewer wants more details, we are happy to provide them.
>
> 4) We are using the Gaussian kernel in our experiments. As suggested by the reviewer we are performing other experiments for different kernels. In particular some preliminary results show that with the Laplacian kernel on the same datasets used in the paper, the same advantages of Park are still valid. We aim to finish further experiments for more kernels and add them in the main paper.
>
> 5) In the setting $Q >> 1$ the cost of the greedy algorithm will weight too much on the overall cost of the algorithm, and the accuracy does not improve further.
>
> 6) The difference in performance between Park-Uni and Park is relative as Park-Uni is never able to match the accuracy of FALKON.
>
>  7) Computing the first principle angle on the dataset would require to at least construct the kernel matrix between all samples which is unfeasible for the large scale regime where our experiments take place.

---

### Official Review · Reviewer_STtd · 2021-07-13

**Rating:** 5
**Confidence:** 4

**Summary:**

The author introduces a partitioning method for solving large-scale kernel ridge regression. Instead of working on the input space, the author proposes constructing partitions directly in the feature space. The effectiveness of the method is demonstrated through numerical experiments. The main limitations are also discussed.

**Limitations And Societal Impact:**

1. The author claims that considering partition in the feature space rather than in the input space allows promoting orthogonality between the local estimators, and can control the local effective dimension and the local bias. However, such a claim is not clearly explained in the rest of the paper. It remains unclear why the space partition can promote orthogonality, and how. In addition, the comparison between space partition and data splitting is lacking, it is better to add the results of the data splitting method in the numerical experiments.

2. In algorithm 1, there are steps on calculating the set $[n]_{\bar q}$. What is the set used for?

3. The proposed method results in a union of local estimators, when predicting, the data needs to be located first based on the partition. Is the final result sensitive to the way of partition? How to deal with the estimation on the boundaries of the partitions?

**Main Review:**

It is an interesting idea considering partition directly in the feature space rather than the input space. However, the advantages are not clearly shown in the work.
The choices of centroids and hyper-parameters are carefully discussed.

**Time Spent Reviewing:**

3

---

> ### Author Response · Authors · 2021-08-10
> **Reply to Reviewer STtd**
>
> We thank the reviewer for their useful feedback. We answer in the following the questions and clarifications presented in the review, and we will add a paragraph in the main paper discussing each one of them
>
> 1)  Propositions 2 and 3 show how the lack of orthogonality of the partition affects the estimation error.  Based on this characterization, we motivate the use of a partitioning scheme which maximizes orthogonality. Also note that our approach allows to resolve important quantities already observed in previous works. Notably, our proposition 3 characterizes what in [30] was only left as an assumption, namely how the sum of local effective dimensions can be bounded by the global effective dimension.
> Concerning the experiments we added Divide and Conquer (D&C) as a baseline for data splitting. Because standard D&C can still not scale on the size of these dataset we consider a version of D&C where each local estimator is computed with FALKON. We run two instance of this D&C with FALKON algorithm: a first one (D&C-Fv1) with all the hyper-parameters chosen with the same scheme that has been used for ParK(-Uni) and FALKON, and a second one (D&C-Fv2) where the Nyström centers are 5 times the value of the previous version (to be sure the approximation is not hurting the performance). What can be seen is that the computational gains of these algorithms come at the cost of a much worse accuracy than ParK(-Uni), which still performs better in time in most cases. In particular these are the numbers in the format DATASET, ERROR, TIME:
>
>  D&C-Fv1: TAXI, $356.2 \pm 0.2$, $14 \pm 1$ - HIGGS $0.212 \pm 0.000$, $50 \pm 1$ - AIRLINE, $0.834 \pm 0.005$, $27 \pm 1$ - AIRLINE-CLS, $33.2 \pm 0.1$, $20 \pm 1$
>
>  D&C-Fv2: TAXI, $327.4 \pm 0.1$, $29 \pm 1$ - HIGGS $0.195 \pm 0.000$, $288 \pm 2$ - AIRLINE, $0.799 \pm 0.005$, $96 \pm 1$ - AIRLINE-CLS, $32.2 \pm 0.1$, $73 \pm 1$
>
> 2) The notation $[n]_q$ is used to denote the set of indices of all sample in partition $q$ (see line 87 for its definition)
>
> 3) Yes the final result is indeed sensitive to the partition as quantified by our analysis in terms of principal angles of the partition. Further, in continuous settings points are unlikely to land exactly on the border of a partition, and the ties can be anyway broken arbitrarily without affecting our analysis(e.g. if a point is on the boundary between the Voronoi cell $q$ and $q'$ the tie can be broken by assigning it to the partition with smaller $q$)

---

### Official Review · Reviewer_J1kM · 2021-07-16

**Rating:** 5
**Confidence:** 4

**Summary:**

The paper proposes a new large-scale solver for kernel ridge regression (KRR). To be specific, the paper first makes a partition on feature space, and in each local space, a KRR estimator with Nystrom subsampling and preconditioner are trained. The paper also presents the theoretical analysis and numerical experiments.

**Limitations And Societal Impact:**

The studied problem and proposed method are not very interesting.

**Main Review:**

I appreciate the solid theoretical proofs, but the proposed method seems just a wrapper of some existing techniques and lacks novelty. Large-scale KRR is also few applied in practice at present.

**Time Spent Reviewing:**

48

---

> ### Author Response · Authors · 2021-08-10
> **Reply to Reviewer J1kM**
>
> 1) We would like to remark that our paper contains several original ideas. The idea of constructing partitions in the feature space is novel. Being based on this new idea, our method is not simply a wrapper of existing techniques. The corresponding analysis, and in particular the characterization of bias and variance of partitioned kernel estimators in terms of the orthogonality of the partition, is also new.
>
> 2) One main point of our work is precisely to promote kernel methods by making them more usable in the modern large-scale scenario. Kernel methods are also an important test bed to try out new ideas for more flexible regression methods (e.g. NTK) or probabilistic models (e.g. Gaussian Processes)

---

> ### Comment · Reviewer_J1kM · 2021-08-28
> **Keep score unchanged**
>
> I have read author's responses and they partially answer my questions, but I believe the current version has not met the standard of NeurIPS, thus I keep my score unchanged.

---

### Official Review · Reviewer_QUZt · 2021-07-19

**Rating:** 7
**Confidence:** 4

**Summary:**

In nonparametric methods such as kernel ridge regression, the number of data is a bottleneck for calculation while large number is needed for the performance. The paper considers the partition of space to reduce the computation time by using the subset of data in each partition that the testing datum belongs to. The authors use the partitioning in the feature space by maximizing the minimal principal angle between partitions, and Determinantal Point Process (DPP) equation is used ensuring the diversity of subspaces.

**Limitations And Societal Impact:**

1. Experiments need more explanation for reproduction.

2. Lack of comparison with other partitioning algorithms


**Main Review:**

The paper is clearly written and coherent with its objective from the motivations to the experiments. The objective is to reduce the computation complexity while preserving the accuracy with many data. In the experiment, the authors showed the result with up to 10^9 number of data, reducing the calculation time significantly while preserving the prediction performance.

The algorithms are reasonable including the choice of DPP equation for selecting centroids.

For minor comments, the authors need more explanations about their experiments to ensure the reproducibility. For example, authors did not provide the source of datasets including the TAXI having 10^9 number of data.

The authors commented that they used GPUs. Explanations have to be provided which part of the algorithm is implemented by GPUs. That is an important explanation because calculation time is an important metric in this work.

The reduction of calculation time varies for different datasets. For TAXI, the reduction is half while it is 1/5 for AIRLINE-CLS. If the authors used commonly the 32 partitions for all datasets, why the time reductions are different? And why the reduction is not 1/32?


**Time Spent Reviewing:**

5

---

> ### Author Response · Authors · 2021-08-10
> **Reply to Reviewer QUZt**
>
> We thank the reviewer for their useful feedback. We answer in the following
> the questions and clarifications presented in the review, and we will add a paragraph in the main paper discussing each one of them
>
> 1) All the pre-processing used in our experiments are the same ones of [19] as explained in line 296, 297
>
> 2) The local FALKON estimators use the same implementation of [19], thus GPUs are used to compute all the possible matrix-vector products and the Cholesky decompositions. Same applies for the greedy algorithm, where GPUs  are used to compute all possible matrix-vector products. We will clarify this in the paper.
>
> 3)  The time reductions are different because not all the Voronoi cells have the same cardinality, so the speedup is not simply a fraction $1 / Q$. We will be more clear about it in the paper.
>
> 4) Concerning the lack of comparison with other methods we decided to add a Divide and Conquer (D&C) baseline. Because standard D&C can still not scale on the size of these dataset we consider a version of D&C where each local estimator is computed with FALKON. We run two instance of this D&C with FALKON algorithm: a first one (D&C-Fv1) with all the hyper-parameters chosen with the same scheme that has been used for ParK(-Uni) and FALKON, and a second one (D&C-Fv2) where the Nyström centers are 5 times the value of the previous version (to be sure the approximation is not hurting the performance). What can be seen is that the computational gains of these algorithms come at the cost of a much worse accuracy than ParK(-Uni), which still performs better in time in most cases. In particular these are the numbers in the format DATASET, ERROR, TIME:
>
>  D&C-Fv1: TAXI, $356.2 \pm 0.2$, $14 \pm 1$ - HIGGS $0.212 \pm 0.000$, $50 \pm 1$ - AIRLINE, $0.834 \pm 0.005$, $27 \pm 1$ - AIRLINE-CLS, $33.2 \pm 0.1$, $20 \pm 1$
>
>  D&C-Fv2: TAXI, $327.4 \pm 0.1$, $29 \pm 1$ - HIGGS $0.195 \pm 0.000$, $288 \pm 2$ - AIRLINE, $0.799 \pm 0.005$, $96 \pm 1$ - AIRLINE-CLS, $32.2 \pm 0.1$, $73 \pm 1$

---

### Decision · Program_Chairs · 2021-09-28

**Decision:**

Accept (Poster)

**Comment:**

The paper proposes an algorithm make the learning of a kernel-based  in the large scale scenario using, as its core, a partition computed in the kernel feature space. If the paper goes beyond a pipeline of existing methods, the paper would gain a lot of strength if:
-  beyond the combination of existing methods, the amount of novelty of each part of this combination was discussed: how new is the feature space partitioning? How would it differ from a simple Kernelized partitioning methods with a clever (greedy) initialization?
- what is the credit assignment in place regarding the proposed pipeline: what makes the method work? The partitioning, the Nystrom approximation? The optimisation procedure? This should be clarified.
- Would other acceleration methods dedicated to kernels such as random features (Rahimi and Recht) would fit into the pipeline? If no, why?

If the technical parts are okay, there is a lack of perspective on the work, preventing the reader to know what are the key parts that make the proposed method effective.

**Consistency Experiment:**

NeurIPS has a long history of experimentation. In 2014, NeurIPS ran an experiment in which 10% of submissions were reviewed by two independent committees to quantify the randomness in the review process. This year, we repeated a variant of this experiment to see how the quality of the review process has changed over time.  This paper was part of the experiment and was therefore assigned to two committees (consisting of reviewers, an Area Chair, and a Senior Area Chair) that reached independent decisions.  If both committees made the same recommendation, this recommendation was followed. If a single committee recommended acceptance, the paper was accepted (with the exception of a few cases in which the other committee identified what we considered a fatal flaw, e.g., an error in a key result).

This copy’s committee reached the following decision: **Reject**

The other committee assigned to the paper recommended **Accept (Poster)**.  You can find the other set of reviews, along with any follow up discussion with the authors here:
https://openreview.net/forum?id=reOnED4N_P-